# Understanding the Implementation of “Sit Less at Work” Interventions in Three Organisations: A Mixed Methods Process Evaluation

**DOI:** 10.3390/ijerph18147361

**Published:** 2021-07-09

**Authors:** Kelly Mackenzie, Elizabeth Such, Paul Norman, Elizabeth Goyder

**Affiliations:** 1School of Health and Related Research, University of Sheffield, Regent Street, Sheffield S1 4DA, UK; e.such@sheffield.ac.uk (E.S.); e.goyder@sheffield.ac.uk (E.G.); 2Department of Psychology, University of Sheffield, Cathedral Court, 1 Vicar Lane, Sheffield S1 2LT, UK; p.norman@sheffield.ac.uk

**Keywords:** workplace, occupational, sitting, sedentary behaviour, process evaluation, organisational culture

## Abstract

Long periods of workplace sitting are associated with poor health outcomes. Interventions to reduce workplace sitting time have had variable impacts, the reasons for which require further investigation. In this paper, we report on a process evaluation aiming to determine the intervention fidelity of three “sit less at work” interventions and to explore barriers and enablers to implementation, using a mixed methods “before and after” intervention study design. Convenience samples of staff were recruited from three diverse organisations to participate in pre- and post-intervention online questionnaires, objective measures of sitting time (using activPAL3™ devices) and post-intervention focus groups. Intervention implementers and key personnel were also recruited to participate in post-intervention focus groups and interviews. The process evaluation found that none of the interventions were implemented as intended, with no consistent reductions in sitting time. Contextual and organisational cultural barriers included workload pressures and the social norms of sitting, competing priorities, lack of management buy-in, and perceptions of where the responsibility for behaviour change should come from. To ensure effective implementation of future initiatives, deeper organisational-level change, requiring buy-in from all levels of management and staff, may be needed to shift organisational culture and associated social norms.

## 1. Introduction

Prolonged sitting time has been shown to be associated with increased risks of health problems such as cardiovascular disease [1,2], type 2 diabetes [2,3], metabolic syndrome [4,5], colon cancer [6] and depression [7]. Furthermore, regularly sitting for long periods of time is associated with an increased risk of all-cause mortality [2,3,8,9,10]. Prolonged sitting behaviour has, therefore, been identified as an important public health concern [11]. Due to the increase in office-based jobs over recent decades [12,13] and the fact that adults spend the majority of their waking hours at work [14,15], the workplace has been identified as a setting where prolonged sitting is a particular concern. Subsequently, there has been an increase in studies evaluating interventions to support staff to sit less during office-based work [16,17,18,19,20,21]; however, a comprehensive systematic review [22] found the effectiveness of such interventions to be mixed.

A recent qualitative systematic review [23] aimed to address existing gaps in the literature by determining the key considerations for the development, implementation and evaluation of interventions to reduce workplace sitting time. The main output of this review was an operational framework to guide intervention development, implementation and evaluation. This framework suggests that the evaluation of interventions to reduce workplace sitting should encompass, and report on, both process and outcome evaluations. Assessing how an intervention was implemented via a process evaluation [24,25] can provide important information to policymakers and practitioners about how to replicate the intervention and what generalisable knowledge can be drawn from the implementation [26,27]; however, studies that have evaluated interventions to reduce workplace sitting have not explicitly mentioned the use of process evaluation and there is a general lack of evidence on the potential barriers to effective implementation of workplace interventions [23]. 

This study is part of a larger project involving the development, implementation and evaluation of interventions to reduce workplace sitting time in organisations of different size and sector (see full project report [28] and related papers [23,29] for more details). This paper covers the process evaluation of three co-produced “sit less at work” interventions, which were implemented in three organisations. The primary aims of this study were to: (1) determine whether the interventions were implemented as intended; (2) explore the barriers and enablers to implementation. A secondary aim was to determine whether the measures used to assess workplace sitting time were feasible and acceptable.

## 2. Materials and Methods

This study utilised a mixed methods “before and after” intervention study design involving pre- and post-intervention online questionnaires and objective measures of sitting time, as well as post-intervention focus groups and interviews. The study was conducted between September 2018 and May 2019. There were three participating organisations: a small business, a charity and a local authority. Detailed descriptions of the participating organisations have been published elsewhere [29].

### 2.1. Interventions

Staff volunteers from each participating organisation, along with a researcher acting as a facilitator, co-produced their own organisation-specific “sit less at work” intervention, resulting in three bespoke interventions. Co-production techniques involved creative thinking workshops and an ecological approach was used to ensure behaviour change determinants at multiple levels of influence were considered [30]. The content of each of the three co-produced interventions can be found in Appendix A. In line with previous studies that have used co-production techniques, communications about the interventions came from senior leaders in the organisations [19,31]. The charity and local authority suggested initiatives consistent with previous studies, such as incorporating standing or moving into team meetings [31,32], replacing periods of sitting with activity (e.g., holding standing or walking meetings) [31] or encouraging lunchtime walks or some other form of exercise [32]. Except for the use of computer prompts [31,32,33], the small business suggested initiatives that were quite different to interventions from previous studies, including press-up and ping pong competitions, doing gym-style exercises with a set of office dumbbells and purchasing wireless headsets to allow staff to move whilst on the phone.

It was the responsibility of each participating organisation to implement their own intervention. Researcher involvement in implementation was limited to enhance external validity. For the small business, individual staff members were assigned actions to implement the intervention. In the charity and local authority, the implementation was the responsibility of named contacts within the human resources team. Further details relating to intervention development, content and plans for implementation can be found in a linked paper also submitted to this Special Issue [34].

### 2.2. Participant Recruitment

There were three parts to the participant recruitment. First, convenience samples of participants were recruited via email from each participating organisation to take part in the “before and after” intervention measures. Second, staff who participated in the “before and after” intervention measures were asked via email to take part in the focus groups. Convenience sampling methods were used rather than probability sampling techniques, as the aim of this research was not to establish intervention effectiveness, but instead to determine how the interventions could be implemented in “real-world” settings. Third, purposeful sampling was used to recruit intervention implementers and key personnel (e.g., human resources, occupational health, managing directors or chief executives) for focus groups and interviews. All participants provided written informed consent and ethical approval for this study was obtained from the School of Health and Related Research Ethics Committee at the University of Sheffield, United Kingdom (ref no. 019368).

### 2.3. Procedures

All “before and after” intervention participants were invited to attend a brief information session held in their workplaces. This session involved a further explanation of the project and a demonstration of how to attach the activPAL3™ device (the inclinometer device used to measure workplace sitting time behaviours) and how to complete the associated logbook. One week prior to the start of the intervention, the “before and after” intervention participants undertook the baseline round of data collection (T0). The interventions were then rolled out over a 12-week period. During the final week of the intervention, a second round of data collection was conducted (T1). Twelve weeks post-intervention marked the final round of data collection (T2). Each round of data collection involved wearing the activPAL3™ device for seven days and completing the logbook and an online questionnaire. Focus groups and interviews lasted 30–60 min and were conducted four to six weeks after the intervention period with a sub-group of the “before and after” intervention participants, the intervention implementers and key personnel. Figure 1 presents a flow chart of the research strategy used in this study.

### 2.4. Data Collection and Analysis

All “before and after” intervention participants completed a pre-intervention questionnaire (at T0), which collected demographic information (age, gender, postcode, ethnicity), educational attainment, job category, less-than-fulltime status and average daily working hours. Online questionnaires completed at T1 and T2 collected data on intervention awareness and safety and adverse events and are reported as frequencies. Participation and drop-out numbers were also collected as part of the process evaluation. Numbers that participated were defined as those who participated in each round of activPAL3™ data collection. Reasons for drop-out or non-participation are also provided where available. Focus groups, conducted with sub-groups of the “before and after” intervention participants, explored issues relating to participant perceptions of the intervention and the outcome measures used, using a semi-structured topic guide to capture six key dimensions (see Table 1). In the charity and local authority settings, semi-structured interviews were conducted with the intervention implementers and key personnel to explore issues relating to intervention implementation (see Table 1 for details of the key dimensions covered). In the small business setting, the “before and after” intervention participants were also the intervention implementers, so a combined focus group was conducted, which explored perceptions of the intervention and thoughts about how the intervention was implemented.

All focus groups and interviews were audio-recorded and transcribed verbatim. Focus group and interview transcripts were uploaded onto NVivo v11 and thematically analysed. Analysis was primarily conducted using pre-defined themes to cover key aspects of the process evaluation [35], which included issues relating to: (1) intervention fidelity; (2) intervention satisfaction and suggested improvements; (3) contextual factors; (4) sustainability of intervention components; (5) outcome measure acceptability. Inductive thematic analysis was also carried out, which allowed for the emergence of additional themes. Once data had been analysed for each organisation, cross-cutting issues and differences between the organisations were explored as part of the inductive analysis.

Data relating to time spent sitting at work, measured objectively using activPAL3™ device worn on the thigh of participants continuously for seven days [32], were collected at T0, T1 and T2. This was accompanied by completion of the daily log (adapted from [36]) to support the interpretation of the activPAL3™ data, where participants indicated the times they woke up, got to work, left work, went to bed and removed the device. These data were collected to provide useful contextual information for understanding the intervention implementation processes, rather than to determine intervention effectiveness. Data for each participant at each time point (T0, T1 and T2) were downloaded from the activPAL3™ devices using software from PAL Technologies, Ltd. The average daily working time and average daily sitting time at work for each data collection period were used to calculate the percentage of daily sitting time during working hours. Outcome data were exported into SPSS v25 and analysed at an organisational level, and are presented as means and standard deviations. These data were also presented as minutes of sitting during an 8-h workday to provide meaningful context. Absolute mean differences were determined by comparing T1 to T0 and T2 to T0 for each organisation for sitting time at work. Relevant paired data for the measure of workplace sitting time were then inputted into the G*Power v3.1 tool (http://www.gpower.hhu.de/) (accessed on 23 March 2019) to compute effect sizes. Cohen’s dz values were used as effect size indices. Cohen defines dz < 0.2, dz = 0.2, dz = 0.5 and dz = 0.8 as “trivial”, “small”, “medium” and “large” effects, respectively [37].

## 3. Results

A total of 57 participants initially volunteered to take part in the “before and after” intervention measures (small business *n* = 5, charity *n* = 11, local authority *n* = 41). Table 2 summarises participant demographic and work-related characteristics and shows that most participants were well-educated, White British women, aged late thirties to early forties. The small business demonstrated some differences compared to the other participating organisations, as participants were predominantly younger men. In terms of work-related characteristics, most participants worked full-time, which was reflected in similar mean daily working times across the three organisations. Participants ranged from senior managers to service occupations, with the local authority displaying the greatest variation in job category.

### 3.1. Quantitative Process Evaluation

The numbers of participants that took part in each round of the activPAL3™ data collection are shown in Appendix A. For each organisation, there was a reduction in the number of participants over time. Reasons for drop-out or non-participation included problems with the activPAL3™ devices not recording data, staff turnover, lack of time to complete the data collection, annual leave commitments and opting out of the data collection. In terms of adverse events, six participants experienced some issues with skin irritation as a result of the dressings used to adhere the activPAL3™ devices to the thigh. These irritations were reported to be very mild and only resulted in one person dropping out of the study. No other health and safety issues were reported.

Data relating to intervention awareness are presented in Appendix A. All participants in the small business and charity and most participants from the local authority were aware of the intervention. In the small business, most participants were also aware of each intervention element. In the charity and local authority, awareness of individual intervention elements was generally low.

### 3.2. Qualitative Process Evaluation

The five pre-determined themes (intervention fidelity, intervention satisfaction and suggested improvements, contextual factors, sustainability of intervention components, outcome measure acceptability) are reviewed in the following sub-sections.

#### 3.2.1. Intervention Fidelity

None of the “sit less at work” interventions were implemented as originally intended. According to participants in the small business, the only elements of the original intervention that were implemented fully were exercises with the office dumbbells and walking to the shops for lunch. The reasons that these initiatives were thought to have worked were because, “they sort of met with our interests” *(participant A, focus group, small business)* and were easily accessible. One element that did not occur was the development and use of computer prompts to remind staff to sit less at work. This was due to underestimating the time it would have taken to develop such a prompt. The remaining intervention elements were partially implemented. There were also some actions that developed during the intervention period, which were not originally planned, such as boxing at work and going to the gym together after work.

In the charity, two elements from the original intervention were fully implemented: the clear office or desk policy (i.e., the centralisation of office equipment and stationary); an initial communication sent to managers informing them about the intervention and how they could support and encourage staff to sit less was implemented. The remaining intervention elements were not implemented. The general feeling about the intervention was that it was not promoted effectively enough and that only those who were participating in the data collection were able to relate to “sit less at work” communications.

In the local authority, none of the elements from the original intervention were implemented fully. An initial communication was placed in the Chief Executive’s weekly column and a message about the intervention was put on the local intranet, but this was a “one-off” and was not maintained during the 12 week intervention period, as was intended. Holding standing meetings did occur within some teams, but this was not consistent across the entire organisation. The remaining intervention elements were not implemented. Posters were put up in meeting rooms and near printers reminding staff to stand or move. This initiative was developed during the intervention period and was not part of the original intervention. Some participants were aware of this extra initiative, but those in customer services roles, who use the meeting rooms and printers less often than other staff, found it unhelpful.

#### 3.2.2. Intervention Satisfaction and Suggested Improvements

Participants from all three organisations highlighted that being involved in the project had raised their awareness of the issue of prolonged sitting at work, which had prompted certain small behaviour changes in individuals, e.g., going to the toilet on a different floor or taking a lunchtime walk. Due to the low-profile nature of the interventions in the charity and local authority, participants were disappointed as they were expecting to see or hear more.

In the small business, the importance of investing more time and energy into the implementation of the intervention at the start to make the intervention become more sustainable was suggested. Furthermore, it was highlighted that small businesses might benefit from a more strategic approach to sitting less at work, which comes from a higher level (such as the central government) with additional support available:
*“I think you could… have a centralised government sit less [initiative] that pops up you know and if you subscribe to it as a thing, it will you know, the app comes up, you know you are given the app… and that would be… much more cost-effective and work on a sort of scaled operation rather than… but it wouldn’t have to be internally driven”*(*managing director, interview, small business*).

Intervention implementers from the charity and local authority reported the importance of having the *“right people around the table” (implementer, interview, charity)* at the start of the process in order to support the development and implementation of the intervention. Getting more buy-in from senior management to support the *“grass roots” (participant A, focus group, charity)* intervention development process was suggested by participants from the charity and local authority. This would help to ensure that intervention implementation was not the responsibility of just one individual (the implementer). In addition, it was felt by some staff in the charity and the local authority that the interventions needed to be more bespoke, possibly down to the team level, as certain teams or directorates, e.g., customer services, faced different challenges than other teams.

#### 3.2.3. Contextual Factors

Workload pressures experienced by participants from all three organisations were reported to affect their participation in the intervention. Furthermore, in the charity and local authority, workload pressures for the implementers limited their ability to give the time needed to ensure the interventions were implemented as intended:
*“The only barrier was workload, that was probably the main thing. Because we are encouraged to do a lot of stuff. We know we’ve got the freedom to do a lot of this, and I think it was just time that we could have spent on implementing correctly.”*(*implementer, interview, charity*).

Difficulty overcoming the social norm of sitting was highlighted as an issue in all three organisations. Even when nudged into standing instead of sitting, the social norm overrides this, as was reported in the local authority when a couple of meeting rooms, which had sit–stand desks in them, were set-up for standing meetings, but *“very quickly, they get lowered and chairs get moved in” (key personnel A, joint interview, local authority)*. In addition, there was the perception that standing meetings are *“a bit weird if you’re used to, you all sit round a table” (participant A, focus group, charity)*. Despite these reported barriers, participants from each organisation provided examples of individuals performing sitting-less behaviours, e.g., managers doing walking one-to-ones, standing meetings within specific teams, incorporating movement into meetings, lunchtime walks, walking up to the next floor to go to the toilet, taking regular breaks.

In the small business, both the staff and the managing director were fully on board and supportive of the intervention. Participants from the small business perceived some form of culture shift, associated with the increased awareness of the issues related to prolonged sitting at work,
*“It’s almost like joking with each that we should all be sitting less so it’s almost like ‘sit less’ has become like the common phrase for us to use”*(*participant B, focus group, small business*).

Competing priorities within the larger organisations were identified as a barrier to intervention rollout. The charity was planning a relocation to a new building for the following year, so this was the focus of much of the organisation. It was felt by the key personnel that this had a detrimental impact on the rollout of the intervention:
*“Because people’s priorities around making changes are linked to the move rather than the here and now… the volume of change is really unique for us so I think it’s got lost if I’m being brutally honest”*(*key personnel, interview, charity*).

In the local authority, other health and wellbeing initiatives, such as those linked to mental health, were competing with the sitting less at work agenda. Furthermore, it was reportedly the norm that health promotion initiatives were promoted as short-lived social marketing campaigns, focused on awareness-raising and signposting to resources rather than investing in longer-term behaviour change strategies.

The existing organisational culture acted as a barrier to implementing the interventions, as reported by charity and local authority staff. For example, there was reported to be a *“culture of meetings” (key personnel A, joint interview, local authority)*, which meant sometimes staff could be in back-to-back meetings all day. This left little time for staff to purposefully sit less due to the difficulties associated with breaking the social norm of sitting in meetings; therefore, *“without the whole organisation actively making a change” (participant C, focus group, charity)*, it made it difficult to change behaviour at an individual level. In addition, there were different barriers reported depending on the team or job role, which impacted the ability of participants to instigate any “sit less at work” actions. For example, customer services staff could only take minimal breaks and had to account for their time, so they were *“very restricted compared to the whole of the council” (participant D, focus group 2, local authority)*.

An issue related to organisational culture specific to the local authority was the idea that the responsibility for sitting less at work should lie with the team managers and the individual, and not come from the corporate level:
*“The argument for me would be well actually, all managers out there, you know are responsible for the health and wellbeing of their teams and then people have an individual responsibility and so some of this is a shift towards how much can we help shape and influence and maybe provide a bit of a framework”*(*key personnel B, joint interview, local authority*).

#### 3.2.4. Sustainability of Intervention Components

In the small business, the use of the dumbbells and staff going to the gym together after work continued after the intervention period ended. In the charity, no initiatives persisted and in the local authority only the prompts near the printers and in the meeting rooms remained.

To support the sustainability of the intervention at an organisational level, it was suggested that certain outcomes or broader impacts be assessed, including the impacts on the health and wellbeing of staff and any associated improvements in productivity. Furthermore, whether the intervention resulted in long-term behaviour change was also felt to be an important sustainability metric from an organisational perspective.

#### 3.2.5. Outcome Measure Acceptability

Participants from all three organisations agreed that the outcome measures used (activPAL3™ devices, logbooks, online questionnaires) were acceptable. Only minor issues were reported, including workload pressures when completing the questionnaires, concerns about accuracy when completing the questionnaires, forgetting to complete the logbook and problems with mild reactions to the dressings used to adhere the activPAL3™ devices onto participants’ thighs.

### 3.3. Measures of Workplace Sitting Time

Table 3 summarises the measures of workplace sitting time at T0, T1 and T2 for the three participating organisations. For the small business, there was a small decrease in sitting time when comparing T0 to T1, but by T2 the proportion of sitting time had increased to greater than it was at baseline. For the charity and local authority, the proportion of time spent sitting increased when comparing T0 to T1, but then at T2 had decreased compared to the baseline finding.

In addition, Table 3 highlights effect sizes relating to sitting at work time. Small effect sizes, in the direction of reductions in sitting time at work, were found for the small business when both T1 and T2 were compared to T0. For the charity, a medium effect size was seen when comparing T0 to the T1, but in the direction of an increase in sitting time at work, then a small effect size in the direction of a reduction in sitting time at work was seen when comparing T0 to T2. For the local authority, there was only a trivial difference when comparing both T0 to T1 and T0 to T2; initially, this was in the direction of an increase in sitting time at work, then in the direction of a reduction in sitting time at work.

## 4. Discussion

Findings from this process evaluation determined that none of the interventions were implemented as intended. This lack of intervention fidelity could explain why no consistent reduction in workplace sitting time was observed and demonstrates the importance of conducting process evaluations [26]. Furthermore, this finding suggests that even when interventions are tailored to the specific needs of the staff (via co-production) and organisational approval is secured, implementation requires further consideration and planning. More extensive researcher involvement in the implementation process could have increased intervention fidelity; however, this then limits external validity, making the researcher part of the intervention, which would not support translation of knowledge into the real world. It is noteworthy to highlight that the outcome measures used were perceived to be feasible and acceptable to participants.

Several contextual barriers to intervention implementation were reported as part of the process evaluation. First, workload pressures (for both participants and implementers) and the social norms that exist in relation to sitting at work were perceived to be important barriers. These findings were also identified in a thematic synthesis of factors perceived to influence the acceptability and feasibility of reducing sitting at work [38], suggesting that these could be common barriers linked to office- or desk-based work [39]. The barrier relating to the “culture of meetings”, where sitting is the norm, was attempted to be overcome by incorporating standing or walking into meetings in the interventions co-produced with the charity and local authority; however, in practice, due to the ingrained social norms of sitting, staff felt uncomfortable participating in or instigating these alternatives to sitting, a finding also reported in previous studies [39,40,41]. Second, both the charity and local authority experienced competing priorities, which meant that the “sit less at work” interventions did not get the attention that they needed to successfully support implementation. Third, participants from the larger organisations reported issues with lack of management and organisational-level buy-in for the interventions. This was compounded by the fact that only one implementer was identified in both the charity and local authority to successfully implement the interventions. A more collaborative approach driven by both managers and staff could have resulted in greater engagement and could have enhanced the organisations’ readiness for change [42,43,44]. Finally, the range of job roles and teams within the larger organisations proved to be a barrier to implementation. Some roles, such as those in customer services, were reported to be much more restricted in terms of having the flexibility and autonomy to take regular breaks from sitting [45]. In order to overcome this, it was suggested that interventions be tailored not only to specific organisations, but also down to the levels of individual teams in larger organisations. This contrasts with the small business, where the managing director reported that they might benefit from a more “top down” strategic approach to sitting less at work.

An identified barrier specific to the local authority related to the idea of individuals taking responsibility for their own behaviours. It was reported by key personnel that changing sitting behaviours at work should primarily be the responsibility of the individual or at least the team managers rather than it being a corporate responsibility. This focus on individual responsibility could have been due to a shift in the organisational culture of local authorities—from the traditional “command and control” leadership style, with a top down hierarchical structure, to more of a self-leadership style, where staff manage their own behaviours to meet the standards and objectives of the organisation [46]; however, the difficulty with self-leadership in this context is that unless there are some higher-level corporate changes to the way the organisation is run, for example addressing the “culture of meetings” and social norms of sitting, it makes it very difficult for individuals to change their sitting behaviours. Organisations need to create a strong implementation climate to enhance staff’s means, motives and opportunity to make use of the intervention [44].

Many of these reported barriers are associated with the broader theme of organisational culture. Organisational culture is described as “a set of collective norms that govern the behaviour of people within the company” [47]. It is this culture that can either promote or undermine attempts to change working practices [41]. A study by Such and Mutrie [41] applied an organisational cultural framework to explore how organisational factors and dynamics impact workplace sitting. This study found that prolonged workplace sitting is a behaviour that is constructed as both an ethos (linked to organisational culture) and a social practice (related to the social norms of sitting) [41]. Furthermore, without a formal policy or strategy, the issue of prolonged workplace sitting is not “problematised”, so the norms of sitting prevail. The findings from this study have highlighted that organisational culture can act to promote the existing social norms of sitting, thereby acting as a significant barrier to workplace sitting interventions [41]. Perceiving sitting as a social practice could support a fuller understanding of the associated social barriers to sitting less at work. For workplace sitting interventions to be implemented successfully, there needs to be an attempt to shift organisational culture, which in turn could help to overcome these socially ingrained barriers. In the small business, a shift in organisational culture was reported, with the phrase “sit less” being frequently used as a reminder to encourage colleagues to sit less at work. Furthermore, participants from the small business maintained some of the intervention initiatives, including using the office dumbbells and going to the gym as a group outside of working hours. Although not formally assessed as part of this study, it was likely that pre-existing friendships between colleagues in the small business, linked to the prevailing culture of the organisation, contributed to these changes [48].

Management support was a key issue, as it led to the perceived permission staff needed to feel able to sit less at work. A recent qualitative study by Chau et al. [49] examined manager and employee perspectives on workplace physical-activity-related policies and practices in organisations from a range of industry sectors. The study demonstrated that the way senior managers influenced perceptions was through their own actions and behaviours, by acting as role models or champions for change at work and by leading by example. This emphasises the importance of gaining management buy-in and engagement during the development and implementation of interventions to reduce workplace sitting time. Furthermore, the study by Chau et al. [49] highlighted tension between where the responsibility for such a behaviour change should lie—with the individual or with the organisation. Chau et al. [49] found that employers are generally moving towards facilitating healthy behaviours and encouraging employees to be more actively involved; however, findings from the current study highlighted that some organisations still perceive staff health behaviours to be down to personal choice. Promoting sitting less (and other health initiatives) at work needs to be a joint endeavour, with responsibility at all levels of management and across the organisation [44,49]. The idea that sitting is a social practice, i.e., being what “work” is for employees with sedentary jobs, seems on the one hand inevitable (sedentary jobs equate to sitting at work) [50,51], but on the other hand a relevant insight when considering what might be needed to make the workplace less sedentary. Changes that involve redefining “work” could support employees to sit less. This idea could explain why sit–stand desks are so frequently selected to form part of an intervention [23], as alternative, lower-cost strategies often promote time away from the desk (e.g., making a drink, going to the bathroom more regularly, visiting colleagues at their desks) [38]; however, the cost of sit–stand desks could be a barrier to uptake for employers [23]. A lower-cost alternative to sit–stand desks is to stand or move during meetings; however, the findings from this study demonstrated that staff felt uncomfortable instigating such strategies due to the ingrained social norms of sitting. This psychological discomfort was also reported in a recent qualitative study by Mansfield et al. [40], which asked office workers to stand in meetings and then explored their experiences. This study found that the feeling of discomfort was primarily due to being perceived to be violating the social norms of sitting and “standing out” from others [40]. In addition, participants stated that despite wanting to stand, they ceased their attempts early in order to conform to explicit or implicit social pressures [40].

### 4.1. Implications for Practice

The findings from this study suggest that there is a need to support organisations differently when implementing interventions to reduce workplace sitting. Rigid protocols that state how and when staff should sit, stand or move should be avoided, and instead an intervention that accommodates barriers experienced by different staff within an organisation needs to be developed [38]. Smaller businesses may require more strategic external support with some intervention elements being more prescriptive, such as the use of applications or standard “sit less” messaging. Larger organisations could benefit from: more bespoke action plans for individual teams or departments, which may experience different barriers and enablers to sitting less at work, and more support in terms of implementation, i.e., having more than one implementer. Furthermore, having more integrated management buy-in and adequate time and human resource investment will help to ensure that the “grass roots” intervention can be adequately operationalised, although this needs to be carefully balanced to not disempower staff [52]. Senior leaders should act as role models for “sit less” initiatives, starting with small incremental changes (e.g., having a standing or movement break as a regular item during team meetings [38]), and organisations should develop a formal policy or strategy that explicitly highlights the importance of sitting less at work. These actions would support shifts in organisational culture and help break the ingrained social norms.

### 4.2. Implications for Future Research

The implementation of interventions to reduce workplace sitting time has been shown in the current study to be complex; therefore, future evaluations of such interventions should include a mixed methods process evaluation, reporting details relating to development and implementation, to support any assessment of intervention effectiveness [53]. The process evaluation should be reported to provide important information about how to replicate the intervention and what generalisable knowledge can be drawn from the implementation [26,27]. Encouragingly, there have been several recent sit less studies published which have reported intervention development or implementation processes [54,55,56,57]. Finally, future research could consider contextual factors that may present as competing priorities and could consider focusing on how to alter organisational culture by challenging the social norms of workplace sitting in order to successfully implement such interventions. This could be achieved by using appropriate theoretical frameworks that allow consideration of organisational culture (e.g., Schein’s model of organisational culture [58]) and sitting as a social practice (e.g., social practice theory [59]).

The findings from this study and from a linked paper also submitted to this Special Issue [34] have been used to refine an operational framework to support the development, implementation and evaluation of interventions to reduce workplace sitting time [23]. This refined version of the framework, as well as a detailed description of the changes and their rationale, is provided elsewhere [28], and is intended for use by both researchers and practitioners to support the future development, implementation and evaluation of sit less at work interventions.

### 4.3. Strengths and Limitations

This study has several strengths. First, the use of both qualitative and quantitative methods as part of a process evaluation supported the interpretation of the measures of workplace sitting time [26]. Second, the fact the interventions were implemented with minimal researcher involvement reflects the real-world nature of this study and helps to enhance the external validity. Third, an objective measure of sitting time was taken from participants using activPAL3™ devices. Taking an objective measure of sitting time has been recommended, as subjective measures alone are associated with reporting and recall biases [60], while the assessment of the feasibility and acceptability of the activPAL3™ devices demonstrated their potential usefulness in future trials.

This study also has some limitations. First, although various measures of workplace sitting time were taken as part of the study, the study design and small sample size precluded a formal testing of the effectiveness of the interventions. Instead, these data were used as part of the process evaluation to consider the extent to which the interventions were implemented. Future appropriately powered randomised control trials are required to establish the intervention’s effectiveness. Encouragingly, data were obtained indicating the acceptability and feasibility of the online questionnaires and the activPAL3™ devices used to measure changes in workplace sitting time. Second, as convenience sampling methods were used, it is likely that there was a degree of selection bias; however, as this study aimed to assess intervention fidelity rather than undertake hypothesis testing, it was felt that the participants would be able to provide useful insights. Third, the number of participants recruited to the study was relatively small, and coupled with the selection bias, this meant that the samples were unlikely to be representative of all staff in the participating organisations. Fourth, a common limitation to qualitative research is the lack of generalisability of the findings. Nonetheless, the qualitative findings across the organisations from this study were broadly in line with a previous review, which assessed the feasibility and acceptability of interventions to reduce workplace sitting [38].

## 5. Conclusions

Process evaluation found that none of the interventions were implemented as intended, which could explain why there was no consistent reduction in sitting time behaviours observed in any of the participating organisations. Despite the use of co-production techniques during intervention development, the interventions still had significant barriers to implementation. Contextual and organisational cultural barriers included workload pressures and the social norms of sitting, competing priorities, lack of management buy-in and perceptions of where the responsibility for behaviour change should come from.

The key findings from this study identified both a need for greater support when implementing interventions to reduce workplace sitting time and the need for a sufficiently supportive organisational culture that can change social norms around sitting at work. To ensure effective implementation of future initiatives, deeper organisational-level change, requiring buy-in from all levels of management and staff, may be needed to shift organisational culture and associated social norms.

## Figures and Tables

**Figure 1 ijerph-18-07361-f001:**
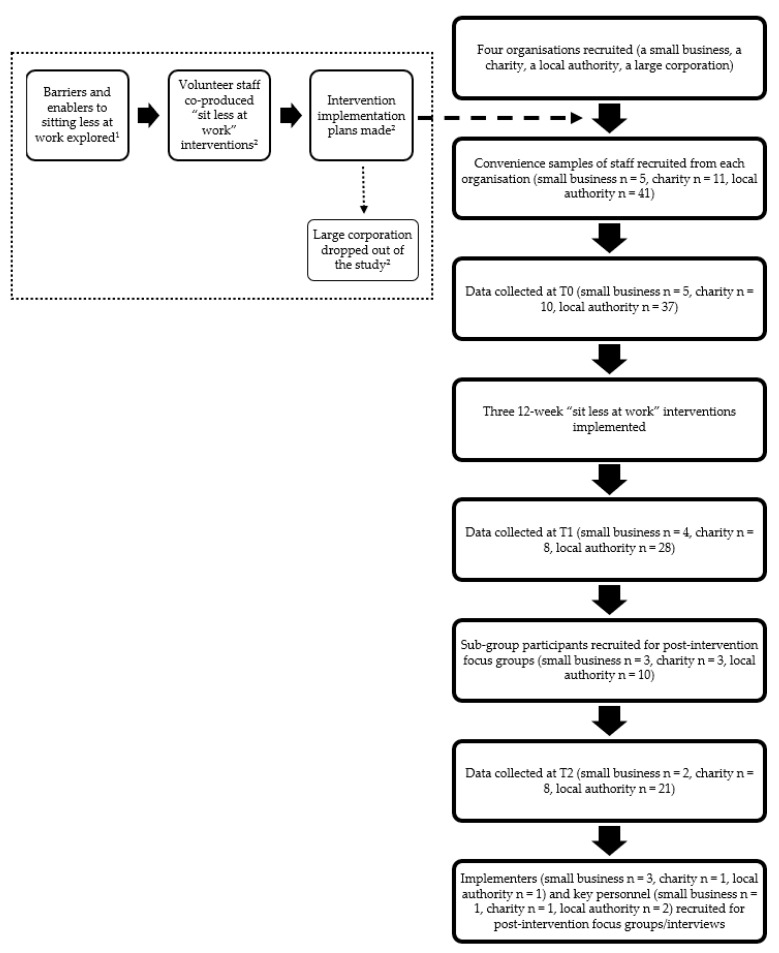
Flow chart of the research strategy. ^1^ Details pertaining to this part of the project are published elsewhere [29]. ^2^ Details pertaining to these parts of the project can be found in a linked paper also submitted to this Special Issue [34].

**Table 1 ijerph-18-07361-t001:** Key dimensions explored during focus groups and interviews.

Key Dimensions Explored during “Before and After” Focus Groups	Key Dimensions Explored during Implementer Focus Groups/Interviews	Key Dimensions Explored during Key Personnel Interviews
1. General perceptions of the intervention as a whole	1. General perceptions about the intervention	1. General perceptions about the intervention
2. Feasibility of the various elements of the intervention and outcome measures	2. The implementation of the intervention	2. The implementation of the intervention
3. Acceptability of the various elements of the intervention and outcome measures	3. Intervention fidelity	3. Feasibility and acceptability of the intervention
4. Elements of the intervention that did/did not work	4. Barriers and enablers to intervention implementation	4. Barriers and enablers to intervention implementation
5. Suggested improvements to the intervention		5. Impacts and sustainability of the intervention
6. Elements or behaviour changes that remained after the intervention		

**Table 2 ijerph-18-07361-t002:** Participant characteristics (those who wore activPAL3™ devices at T0).

Characteristics of Participants at Baseline	Small Business	Charity	Local Authority	Total [%] /Mean
**Total number of staff**	8	488	4146	-
**Total number of participants**	5	10	37	52
**Mean age (years)**	36	38	41	38
**Women (*n*)**	1	7	26	34 [65%]
**Ethnicity**				
- White British (*n*)	5	9	32	46 [88%]
- Other (*n*)	0	1	5	6 [12%]
**Highest educational attainment ***				
- Degree or equivalent (*n*)	1	7	15	23 [45%]
- Higher education (*n*)	2	0	11	13 [25%]
- A level or equivalent (*n*)	2	2	7	11 [22%]
- GCSEs grade A *-C or equivalent (*n*)	0	1	3	4 [8%]
**Full-time (*n*)**	4	7	32	43 [83%]
**Mean daily working time (mins)**	499	471	475	482
**Job Category**				
- Executive, administrator, senior manager (e.g., CEO, Chief Executive, sales manager)	2	2	2	6 [12%]
- Professional (e.g., engineer, accountant, systems analyst) (*n*)	0	4	21	25 [48%]
- Technical support (e.g., lab technician, legal assistant, computer programmer) (*n*)	2	0	4	6 [12%]
- Clerical and administrative support (e.g., secretary, PA, billing clerk, office supervisor) (*n*)	1	4	9	14 [27%]
- Service occupation (e.g., security officer, food service worker, cleaner) (*n*)	0	0	1	1 [2%]

* One non-responder to this question in the local authority.

**Table 3 ijerph-18-07361-t003:** Objective sitting time at T0, T1 and T2 from activPAL3™ data.

Measure	Small Business (*n* = 5)	Charity (*n* = 10)	Local Authority (*n* = 37)
**% sitting time at work at T0, mean (SD)** **[time in minutes/8 h working day]**	81.62 (10.60) [392]	75.29 (9.27) [361]	71.62 (13.83) [344]
**% sitting time at work at T1, mean (SD)** **[time in minutes/8 h working day]**	77.30 (6.55) [371]	76.74 (6.22) [368]	73.27 (13.68) [352]
**% sitting time at work at T2, mean (SD)** **[time in minutes/8 h working day]**	82.94 (7.18) [398]	71.58 (14.69) [344]	67.77 (15.54) [325]
**Mean difference between T1 and T0, % sitting time *** **[time in minutes/8 h working day]**	−4.32 [-21]	1.45 [7]	1.65 [8]
**Mean difference between T2 and T0, % sitting time *** **[time in minutes/8 h working day]**	1.32 [6]	−3.71 [-18]	−3.85 [18]
**Effect size (dz) comparing T1 and T0 ****	−0.38	0.59	0.10
**Effect size (dz) comparing T2 and T0 ****	−0.25	−0.39	−0.07

Values reported are means (standard deviations) from all available data. * Mean differences were determined by subtracting data for T1 or T2 from data for T0. ** Effect sizes were calculated from paired data only.

## Data Availability

The datasets used or analysed during the current study are available from the corresponding author on reasonable request or can be found in the full PhD project report at https://etheses.whiterose.ac.uk/28889/ (accessed on 6 June 2021).

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
