# Peer review of "Understanding the Implementation of “Sit Less at Work” Interventions in Three Organisations: A Mixed Methods Process Evaluation"

_ijerph, 2021, doi:10.3390/ijerph18147361_

Round 1

Reviewer 1 Report

Understanding the implementation of “Sit less at Work” interventions in three organisations: a mixed methods process evaluation

Reference ijerph - 1258391

Comments to Authors and Editor

General comments

The manuscript “Understanding the implementation of “Sit less at Work” interventions in three organisations: a mixed methods process evaluationis not currently ready to publish in IJERPH. The study is completely descriptive in nature. Although the qualitative part is strong enough, the major weakness of quantitative part is scientific validity. I couldn’t find any scientific quality for the quantitative study. I strongly request you please exclude the quantitative part and resubmit the study as a qualitative study if Editors agree.  

Sampling: The selection bias is obvious for convenience sample. I am not sure what is the rationale of using convenience samples to recruit participants. I don’t think any barrier to recruit participant with a probability sampling procedure in a scientific manner.

Study Design: The pre-post design is not an accurate and appropriate for intervention study. The comparator/control group will strength the study significantly.

Sample size and power: I am little concern about 52 participants (in total) to reach an optimum sample size and power of the study, specifically for only 6 and 12 participants for small business and charity respectively. Definitely participants are heterogenous in nature in regard to sitting time and other confounding measures.

Statistical analysis plan (SAP): I couldn’t find any statistical plan for the quantitative study.  Although addressed mean differences between time points, there is no statistical justification whether difference is significant or not. Consequently, intervention works or not.

Results: The results were described by absolute numbers. If Authors describe their results in relative numbers in addition to absolute numbers, the finding would be clearer and more meaningful.

Interventions: What is the rationale of collecting data within the short interval, one in the final week (T1) and the other one was post 12-week intervention program (T2). Any significant changes over the short interval. Rather I prefer 6 months follow-up whether intervention works or not.

Reviewer 2 Report

This paper is very interesting and well-designed by experts in the field. 

What is missing is a flowchart of the strategy and approach followed by the authors.

Further, the discussion is a little weak and I suggest to the authors to further discuss and compare to the large literature in the field.

In terms of data analysis, I am wondering if the authors can perform multivariate analyses to further discriminate and characterizer their groups.

In terms of future recommendations and prospects, the authors failed to propose robust things. Please amend as much as possible.

Reviewer 3 Report

Thank-you for this interesting article presenting a process evaluation using a mixed methods assessment approach to determine the effectiveness, as well as barriers and enablers of a “Sit Less at Work” intervention. Quantitative and qualitative data was obtained and showed that none of the interventions were implemented as intended, and there was no significant reduction in sitting time in 2/3 study sites at 12 weeks, and 1/3 study sites at 24 weeks. Barriers to the “Sit Less” intervention included, contextual and organizational cultural barriers. This work provides information that could be useful for quality improvement of initiatives related to decreasing workplace sedentary behaviours. Below are some suggestions and comments for consideration regarding the manuscript.

General Comments:

As the “Sit Less” interventions were different at each site, were the specific components of each intervention created in collaboration with the workplace where they were implemented or were they determined by the researchers/an outside source? And how do they compare with components and structure of other interventions aimed at workplace movement/reduce sitting time that have been studied.

Specific Comments:

Page 4, Line 151 (Table S4). How many participants were originally contacted to participate? Is it possible to present a flow of participants so it may be known what percentage of people agreed to participate (i.e., the n=57) in the intervention leading to the 52 participants that eventually provided information for the “before and after” study along with the reasons for drop-out or non-participation where available (as noted was collected in lines 110-111)?

Page 7, Line 301: In the small business, given the small sample size, were there friendships outside of the workplace already present among these individuals prior to the intervention being implemented, which could have potentially led to the greater likelihood to participate in active group activities outside of the workplace together (such as going to the gym together) or did this form due to the intervention?

Page 8, Lines 321-323, and Table 3. Was the relocation of the Charity to a new building noted on Page 7, Line 269 accounted for in the analysis of % sitting time, as relocation may be associated with greater movement/less sitting? If so, how was this accounted for?

Page 8, Table 3. The number of participants noted in this table differ from the number of participants noted in Table 2 presenting the participant characteristics, even though both tables note that they relate to the participants who wore the activPAL3TM devices. Why this discrepancy? Also, the number of participants noted in Table 3 appears to be greater than the noted “total of 57 participants initially volunteered” (Page 4, Line 151). Please clarify.

The following is not directly noted in the present manuscript, but relates to how this paper fits within the broader context of the body of research related to “Sit Less”:

In one of the other related referenced studies 4 sites were assessed, the 3 mentioned in the present paper as well as a large corporation. Information from a larger corporation as it pertains to this intervention may also be pertinent. What was the reason for why the large corporation did not participate, or was not included, in the present study?

Thank-you for your consideration.

Round 2

Reviewer 1 Report

Understanding the implementation of “Sit less at Work” interventions in three organisations: a mixed methods process evaluation

Reference IJERPH 1258391R

Comments to Authors and Editor

General comments

The manuscripts improved a lot after consideration of comments and suggestions. However,

I was not clear why mixed method has been dropped off from the study. Pre- and post-intervention online questionnaire and focus group interview were clearly addressed in the abstract and elsewhere, which clearly showed a mixed method approach, if I am not wrong. 
